# Control of Brown Rot Produced by *Monilinia fructicola* in Peaches Using a Full-Spectrum Extract of *Zuccagnia punctata* Cav.

**Melina G. Di Liberto** [1] [iD], **María Inés Stegmayer** [2], **Laura N. Fernández** [2], **Ariel D. Quiroga** [3] [iD], **Laura A. Svetaz** [1,*] [iD] and **Marcos G. Derita** [1,2,*] [iD]

1 Farmacognosia, Facultad de Ciencias Bioquímicas y Farmacéuticas, Universidad Nacional de Rosario, Suipacha 531, Rosario S2002LRK, Argentina; mdiliberto@fbioyf.unr.edu.ar
2 ICiAgro Litoral, Universidad Nacional del Litoral, CONICET, Kreder 2805, Esperanza 3080HOF, Argentina; mistegmayer@gmail.com (M.I.S.); laurafernandez1@gmail.com (L.N.F.)
3 IFISE, Universidad Nacional de Rosario, CONICET, Suipacha 531, Rosario S2002LRK, Argentina; quiroga@ifise-conicet.gov.ar
\* Correspondence: lsvetaz@fbioyf.unr.edu.ar (L.A.S.); mderita@fbioyf.unr.edu.ar (M.G.D.); Tel.: +54-9341-153-061322 (L.A.S.); +54-9341-155-317769 (M.G.D.)

**Abstract:** Brown rot of stone fruit, caused by *Monilinia* spp., is one of the most important diseases worldwide, causing significant production losses. Currently, the standard practices for controlling this infection consist of repetitive use of synthetic fungicides. The global tendency encourages the demand for high-quality food products harmless to health and the environment, leading to a reduction in the use of these types of substances. *Zuccagnia punctata* (Fabaceae) is a perennial shrub extensively used for the treatment of fungal and bacterial infections in Argentinean traditional medicine. In this study, we isolated and characterized (morphologically and molecularly) a pathogenic and virulent strain of *Monilinia fructicola*, which is the most hostile species of the genus. Consequently, we explored the in vitro antifungal activity of the ethanolic extract of *Z. punctata* against this phytopathogen. The chalcones 2′,4′-dihydroxy-3′-methoxychalcone and 2′,4′-dihydroxychalcone were isolated from the extract and evaluated against *M. fructicola* demonstrating that they were responsible for this activity. To promote full-spectrum extract rather than pure compounds, we performed ex-vivo assays using fresh peaches inoculated with the pathogen, and then treated by immersion in an extract solution of 250 μg/mL concentration. Treatment with *Z. punctata* extract did not show a statistically significant difference from commercial fungicides in the control of fruit rot. In addition, Huh7 cell cytotoxicity evaluation showed that *Z. punctata* extract was less cytotoxic than commercial fungicides at the assayed concentrations. Based on our research, this plant extract could potentially offer a safer alternative to commercial fungicides for treating peach brown rot.

**Keywords:** phytopathogenic fungi; *Zuccagnia punctata*; antifungal; brown rot; *Monilinia*; peach; chalcones

## 1. Introduction

*Monilinia fructicola*, the cause of brown rot, is the most critical phytopathogenic fungus producing severe stone fruit production losses worldwide, with high economic costs during both the growing and post-harvest stages. Apart from *M. fructicola*, there are two other species that produce brown rot: *M. laxa* and *M. fructigena*, both less important in terms of incidence. *M. fructicola* is the most aggressive species in the genus and is of particular concern due to its ability to undergo sexual recombination, which can lead to the development and establishment of fungicide-resistant strains [1]. In addition to peaches, cherries, and plums, pome fruits such as apples and pears are also affected.

In the field, *Monilinia* causes blossom blight and twig cankers, producing direct losses and spreading inoculum on the surface of fruits [2,3]. These field infections are latent and

develop during the postharvest period of shelf life, mainly when the temperature is optimal for its development. Its management relies on an integrated strategy that includes orchard fungicide control, local practices, and optimization of storage conditions [4]. The extended susceptibility period from flowers to fruits requires several fungicide applications resulting in chemical residues on fruit surfaces [5]. In many countries, no postharvest chemical treatments are allowed for stone fruits, emphasizing the need for alternative post-harvest treatments.

In Argentina, only *M. fructicola* and *M. laxa* have been reported, although *M. fructicola* prevails in most stone fruit-producing regions. Even in places like Mendoza and the Upper Valley of Río Negro and Neuquén, where climatic conditions are not so favorable for disease development, *M. fructicola* significantly affects the genus *Prunus* [6]. Moreover, *Monilinia* is a global pathogen with a high potential to spread to other countries and hosts [7–9]. In 2016, brown rot was first reported in local cold storage pears in Argentina [10].

In addition to reduced crop yields, product quality and safety are also compromised. Tightened international regulations and public concerns demand safer alternative fungicides, such as new extracts [11], essential oils [12], and natural compounds [13], as well as the design of semi-synthetic derivatives [14] and novel compounds from natural sources [15]. A study in Argentina explored native plants and their extracts and found that some showed strong antifungal activity against *M. fructicola*, suggesting the potential of crude extracts as new fungicide sources [16].

*Zuccagnia punctata* Cav. (Fabaceae) is a monotypic South American shrub widely distributed in arid and semiarid areas of western Argentina [17]. Commonly known as jarilla, jarilla macho, pispito, and pus–pus [18], this species was chosen for the evaluation of its traditional use as a foot antiseptic and for the treatment of bacterial and fungal infections, asthma, arthritis, rheumatism, and tumors [19,20]. Numerous studies on its antifungal activity against yeasts, filamentous fungi, and Basidiomycetes have been conducted since the first report in 2001 by Quiroga et al. (2001) [21–25]. Our research group has been conducting phytochemical and antifungal activity studies of this species for 20 years [26–30], including the combination of extracts with others from native species to enhance their activity and decrease their active concentration [31]. The plant, its extracts, isolated compounds, and phytotherapeutic preparations have been widely reported in the literature [32–40], but this is the first report of ex vivo fruit control against phytopathogenic fungi.

The aim of this study was to isolate and characterize the *M. fructicola* strain obtained from peaches that presented the corresponding signs and symptoms of brown rot. Additionally, we aimed to evaluate the in vitro antifungal activity of the ethanolic extract obtained from the aerial parts of *Z. punctata* (*Zpu*E), and isolate and identify its main active compounds. Furthermore, a test was carried out using a phytochemically characterized extract on post-harvested peaches. Finally, the cytotoxicity of the extract on human hepatoma cells was determined and compared to that of commercially used fungicides.

## 2. Materials and Methods

### 2.1. Plant Material and Extract Preparation

Aerial parts of *Z. punctata* were collected in March 2016 at the intersection of national route RN147 and provincial route RP26 (Ayacucho, Depto. Belgrano, San Luis Province). The plants were identified by Dr. Martin Hadad, Instituto de Biotecnología (IBT), and a *voucher specimen* was deposited at the Herbarium of the National University of San Luis (UNSL) [Ejército de los Andes 950 (D5700HHW) San Luis, Argentina, code: UNSL Del Vitto & Petenatti 9230].

Air-dried aerial parts (100 g) were powdered using a Fritsch Pulverisette-15 mill (Germany). The material was macerated (3 × 24 h each) with ethanol (EtOH) under mechanical stirring using a Heidolph RZR 50 apparatus (Schwabach, Germany). After filtration and evaporation, an EtOH crude extract was obtained (27.71 $w/w$ in terms of dry starting material).

### 2.2. Isolation and Chemical Characterization of Natural Compounds

Compounds 2′,4′-dihydroxy-3′-methoxychalcone (**1**) and 2′,4′-dihydroxychalcone (**2**) were isolated from *Zpu*E according to a previously reported procedure [41]. Compounds **1** (187 mg, 0.24% *w/w* in terms of dry starting material) and **2** (97 mg, 0.12% *w/w* in terms of dry starting material), were identified by micro-melting point, optical rotation, and spectroscopic data, and were compared with authentic samples obtained previously in our laboratory [26,27,29]. [1]HNMR, [13]CNMR, and mass spectra are presented in the Supplementary Material (Figures S1–S6).

### 2.3. Antifungal Studies

#### 2.3.1. Microorganisms and Media

A monosporic isolate of *Monilinia* was obtained from peaches that presented the typical symptoms of brown rot. The isolate was incubated in a Potato-Dextrose-Agar (PDA) medium at $25 \pm 1$ °C with a 12 h photoperiod for 7 days, and the morphological characteristics were evaluated. For future use, the isolate was conserved at the Institute of Agricultural Technology (INTA, EE San Pedro, Argentina) under the code INTA-SP345 and conserved at $-20$ °C on dried filter paper in the mycological collection of ICiAgro Litoral, UNL, CONICET, FCA, Argentina. The inoculum for bioassays was obtained according to the Clinical and Laboratory Standards Institute reported procedures and adjusted to $1 \times 10^4$ Colony Forming Units (CFU)/mL [42].

#### 2.3.2. Molecular Characterization

The identity of the *Monilinia* isolate was confirmed by molecular characterization. Fungal genomic DNA was extracted from 7-day-old cultures grown on PDA at $25 \pm 1$ °C, as described by Gupta et al. (2013) [43], and was used as the template for PCR amplification of a segment of the ITS (Internal Transcribed Spacer) region of ribosomal nuclear DNA (rDNA) using the primers ITS4 (5′-TCC TCC GCT TAT TGA TAT GC-3′) and ITS5 (5′-GGA AGT AAA AGT CGT AAC AAG G-3′) [44]. PCR reactions were performed on a Techne TC-312 thermal cycler (Techne, Cambridge, UK) in 20-µL reaction mixtures containing $1\times$ PCR buffer, 2.5 mM MgCl$_2$, 0.4 µM each primer, 0.2 mM dNTPs, 1 U of Taq DNA polymerase (PB-L, Productos Bio-Lógicos®, Rosario, Argentina), and 100 ng of genomic DNA. Amplifications were programmed to carry out an initial denaturation step at 94 °C for 5 min, followed by 36 cycles, each consisting of a denaturation step at 94 °C for 30 s, an annealing step, and an extension step at 57 °C for 30 s and at 72 °C for 30 s. The final extension was carried out at 72 °C for 7 min. PCR products were visualized under ultraviolet light on 1.5% (*w/v*) agarose gel in $1\times$ TAE buffer stained with GelRed (Biotium, Hayward, CA, USA). A UST-30M-8E Biostep transilluminator (Biostep, Jahnsdorf, Germany) was used. The amplified products were purified and sequenced using the same primers from Macrogen (Seoul, Korea). Identification was performed by comparing the sequences with all fungal sequences from the GenBank Nucleotide Database hosted by the National Center for Biotechnology Information (NCBI, https://www.ncbi.nlm.nih.gov/ accessed on 26 May 2023) using BLAST [45]. The sequences generated in this study were deposited in the GenBank database (accession number OR063822).

#### 2.3.3. In Vitro Susceptibility Tests

The Minimum Inhibitory Concentration (MIC) values were determined using broth microdilution techniques according to CLSI guidelines [42] for filamentous fungi (document M38-Ed3). The 96-well plates were incubated in a moist dark chamber at 20–25 °C, MIC values were visually documented according to the fungus growth control. For the assay, 50 mg/mL or 12.5 mg/mL solutions of each extract/compound were prepared in DMSO. Aliquots of 40 µL were dissolved in 960 µL of Sabouraud dextrose broth to obtain stock solutions and then, serially diluted in medium from 1000 to 3.95 µg/mL for extracts and 250–0.49 µg/mL for compounds (final volume = 100 µL) in each corresponding well of the microtiter trays. Then, 100 µL of fungal spore suspension was added to each well, except

for the sterility control, where sterile water was added instead. The commercial antifungal agents Carbendazim (Cbz) and Captan (Cap) were used as positive controls. The MIC endpoints were visually assigned at the lowest concentration of extract/compound, leading to the total inhibition of fungal growth compared with the growth in the control wells in the absence of an antifungal agent. This assay was performed in duplicate.

The Minimum Fungicidal Concentration (MFC), defined as the lowest concentration of extract/compound that completely kills the fungi, was determined after assessing the corresponding MIC data by transferring a sample aliquot (5 µL) of each clear microtiter well onto a 150-mm PDA plate. Inoculated plates were incubated at 20–25 °C and the MFC values were visually determined after 7–10 days, in accordance with the corresponding growth control.

For the bioautography assay, samples (50 µg of *Zpu*E and 15 µg of each pure compound **1** and **2**) were line-seeded on TLC plates and developed using a mixture of CHCl$_3$:EtOAc (6:4) as mobile phase. The chromatogram was then deposited on a sterile Petri dish lid and the inoculated medium was spread evenly over the surface at a rate of 0.2 mL per cm$^2$. The agar layer was prepared with PD culture medium, 0.6% agar, and 0.02% phenol red, and the final concentration of *M. fructicola* was quantified at $1 \times 10^4$ CFU/mL, according to reported procedures [46]. The inoculated TLC plate was incubated at 20 °C for 6 days. Then, the bioautogram was sprayed with methyl thiazolyl tetrazolium chloride (MTT) (1 mg/mL in sterile water) and incubated for 3 h at 20 °C. Inhibition zones appeared as dark yellow spots on a dark brown background [47].

### 2.3.4. Ex Vivo Antifungal Assay on Wounded Fruits
Peaches Collection and Preparation

Fresh ripe peaches (*Prunus persica* L.) cultivar 'Red Globe' from the *Campo Experimental de Cultivos Intensivos y Forestales* (Facultad de Ciencias Agrarias, Universidad Nacional del Litoral, Esperanza, Argentina) cultivated according to the agroclimatic features of the region [summer temperatures between 25 to 35 °C and 30 to 40 relative humidity (RH)] were selected on the basis of size and absence of physical lesions or symptoms of infection. The fruits were harvested in the absence of any chemical treatment and their surface was disinfected with 2% (*w/v*) sodium hypochlorite for 2 min, then rinsed with tap water and air-dried. The fruits exposed to different treatments were placed in suitable plastic boxes (300 mm × 500 mm × 100 mm) containing filter papers embedded with 25 mL of sterile water at the bottom to maintain high RH (90–95%).

Test on Peaches: Inoculation and Treatment Applications

The test was performed according to the literature [48]. Thirty fruits were randomly assigned into 3 groups of 10 units (test, comparator, and control). The upper zone of each peach was wound (2 mm) with a sterile tip and inoculated with 10 µL of a conidial suspension ($1 \times 10^5$ CFU/mL) of *M. fructicola*. After 2 h, the fruits were individually immersed for 3 s in the different treatment solutions contained in suitable containers. Based on the results of the in vitro antifungal assay, the first group was treated with *Zpu*E (250 µg/mL), and the second group, which acted as positive control and comparator, was immersed in commercial Cbz (1.9 µg/mL). The last group constituted the negative control in which peaches were immersed in sterile water.

Test Development and Evaluation

Inoculated fruits subjected to the different treatments were box-stored for 10 days at 20 °C. At the end of storage, the percentage of infected fruit and the sporulation index [49] of *M. fructicola* were determined for each fruit using a rating index of 0–4, where 0 = no sign of disease, 1 = lesion visible but no sporulation, 2 = sporulating area on lesion smaller than a quarter of the fruit, 3 = sporulating area larger than a quarter of the fruit, but less than half of the fruit; and 4 = sporulating area larger than half of the fruit. The index

value for each unit was treated as a replicate and the treatment means were subjected to statistical analysis.

Statistical Analysis

The experimental data were analyzed statistically by a non-parametric test followed by the Kruskal-Wallis comparison test ($\alpha = 0.05$) using GraphPad Prism v.7.0 software.

### 2.4. Cell Viability Assay

Cell viability test was estimated by the MTT assay [50]. Human hepatoma cells (Huh7) were 24 h treated with *Zpu*E, Cbz, or Cap in different concentrations ($2\times$, $1\times$, $1/2\times$, and $1/4\times$ MIC). The experiment was performed in sextuplicate, and means with standard deviation were calculated. DMSO (10%) was considered 100% death, and all the dilution readings were regarded as 0% viability (positive control). Instead, 0.1% DMSO was considered 0% death, and the reading for all the dilutions was considered 100% viable (negative control).

## 3. Results

### 3.1. Fungal Isolation and Characterization

The fungal isolate was morphologically identified by INTA (EE San Pedro, Argentina) as *M. fructicola* (INTA-SP345). Figure 1a shows the typical symptoms and signs of *Monilinia* on peaches, which are characterized by concentric circular spots composed of masses of gray spores surrounded by darkened tissue, Figure 1b shows the morphological characteristics of the colonies on PDA plates, and Figure 1c shows the features of their characteristic lemon-shaped conidia in chains under the microscope. Figure 1c also shows a conidiophore of microconidia (spermatia), which are also produced in chains but are non-germinative [51].

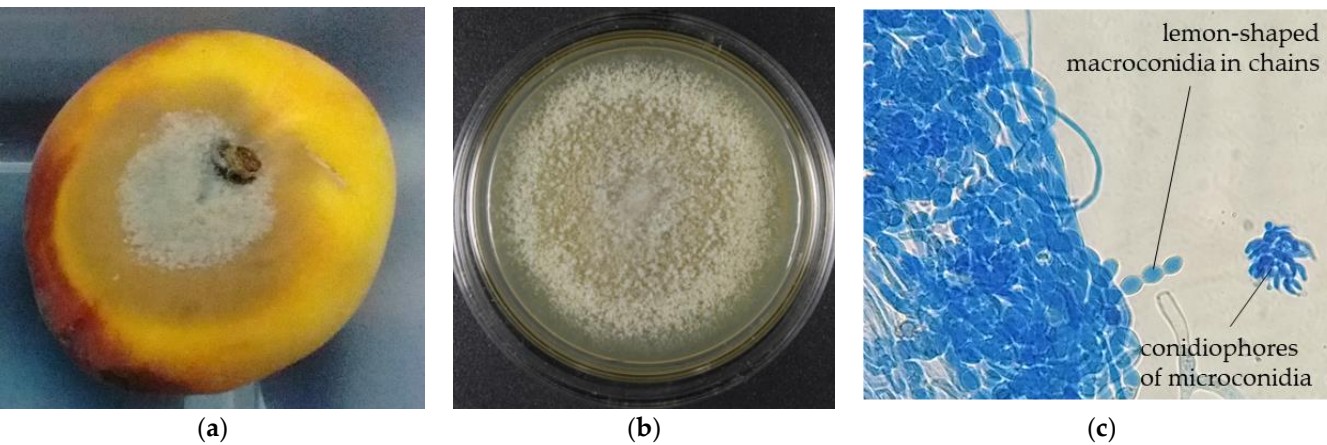

　　(**a**)　　　　　　　　　　　　　　　　(**b**)　　　　　　　　　　　　　　　　(**c**)

**Figure 1.** (**a**) Typical symptoms and signs of *M. fructicola* on peaches. (**b**) Morphological characteristics of colonies developed on PDA after isolation: yellow-grey colonies forming concentric rings. (**c**) Microscopic features of the mycelium.

Molecular identification corroborated this result. The ITS 4–5 regions were amplified and sequenced. A BLAST search showed that the sequence data of the isolated strain IN-TA-SP345 shared 99.9% similarity with *M. fructicola* (EU131181).

### 3.2. In Vitro Antifungal Activity of ZpuE and Compounds **1** and **2** against the Selected Phytopathogenic Fungi

*Zpu*E was evaluated for antifungal activity against *M. fructicola* by using a micro broth dilution assay. In addition, two pure compounds were isolated and assayed against this pathogen. MICs and MFCs of the extract and compounds are shown in Table 1. The chemical structures of the pure compounds are shown in Figure 2.

**Table 1.** Minimum Inhibitory Concentrations (MICs) and Minimum Fungicidal Concentrations (MFCs) (μg/mL) of the ethanolic extract and pure compounds isolated from *Z. punctata* against *M. fructicola* [1].

| Sample | MICs (μg/mL) | MFCs (μg/mL) |
| --- | --- | --- |
| *Zpu*E | 250 | 250 |
| **1** | 62.5 | 125 |
| **2** | 125 | 125 |
| Cbz | 0.97 | 0.97 |
| Cap | 1.9 | 1.9 |

[1] *Monilinia fructicola* INTA-SP345. The commercial fungicides Carbendazim (Cbz) and Captan (Cap) were used as positive controls. *Zpu*E: ethanolic extract of aerial parts of *Zuccagnia punctata*, **1**: 2′,4′-dihydroxy-3′-methoxychalcone, **2**: 2′,4′-dihydroxychalcone. I: inactive (MIC or MFC > 1000 μg/mL for extracts or >250 μg/mL for pure compounds).

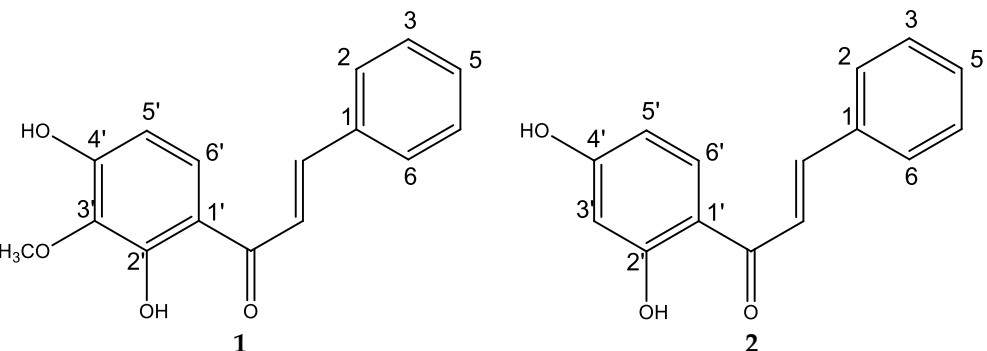

**Figure 2.** Chemical structures of compounds isolated from EtOH extract of aerial parts of *Z. punctata*. (**1**) 2′,4′-dihydroxy-3′-methoxychalcone; (**2**) 2′,4′-dihydroxychalcone.

According to Holetz et al. (2002) [52], *Zpu*E showed moderate fungicidal activity against *M. fructicola*.

Regarding the pure compounds, **1** and **2**, they showed similar activities, displaying MICs = 62.5 and 125 μg/mL, respectively, and MFCs = 125 μg/mL.

*3.3. Detection of Bioactive Compounds by Bioautography and Identification of the Main Active Constituents in the Extract*

*Zpu*E, in addition to pure compounds **1** and **2** as bioactive markers, were developed on a TLC plate and bioautographed using *M. fructicola* as the test microorganism (Figure 3). A replicate of the TLC plate was exposed to UV light at 254 and 365 nm and sprayed with *p*-anisaldehyde sulfuric to chemically reveal the presence of compounds **1** and **2** within the extract. Results showed that compounds **1** and **2** were present in *Zpu*E. Therefore, the bioactivity observed in this extract was principally attributed to the content of these two active compounds. There was at least another active compound in the ethanolic extract as can be seen on the TLC plate with Rf = 0.28, but its identity could not be elucidated due to the low yield obtained. We hypothesized that the activity corresponding to this spot could be attributed to the active flavonoids present in this plant, based on the literature and data obtained in our previous studies against human fungal pathogens [26,29].

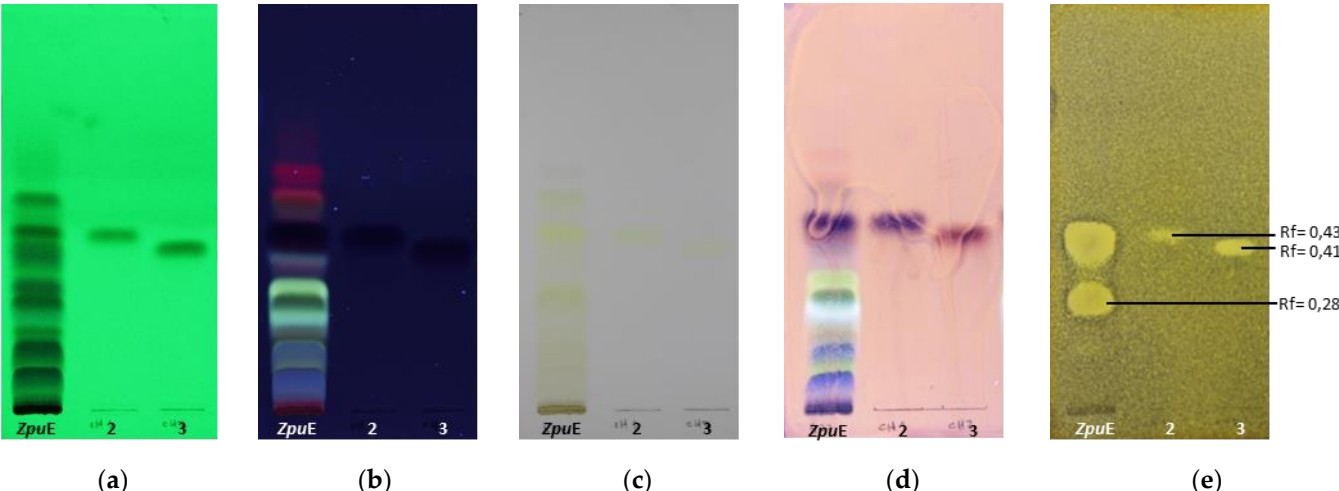

**Figure 3.** TLC developed for *Z. punctata* ethanolic extract (*Zpu*E), and the pure compounds 2′,4′-dihydroxy-3′-methoxychalcone (**1**) and 2′,4′-dihydroxychalcone (**2**). (**a**) Exposed to UV light 254 nm; (**b**) exposed to UV light 365 nm; (**c**) sprayed with *p*-anisaldehyde sulfuric; (**d**) sprayed with *p*-anisaldehyde sulfuric and exposed to UV light 365 nm; (**e**) bioautography using *M. fructicola* as the test microorganism. Mobile phase: CHCl$_3$:EtOAc (6:4).

### 3.4. Effect of Ethanolic Extract of Z. punctata for the Control of Peach Brown Rot on Fresh Fruits

In order to evaluate the effectiveness of *Zpu*E in controlling postharvest disease caused by *M. fructicola*, fresh peaches cv. 'Red Globe' wound-inoculated with the phytopathogen were immersed in a 250 μg/mL solution of the bioactive extract and the sporulation index of infected fruits was determined (see Section Test Development and Evaluation). Commercial product Cbz (0.97 μg/mL) and a control set consisting of sterile water were also tested. After 10 days, the peaches of the control batch were all completely infected and needed to be discarded. Additionally, treated batches with *Zpu*E and Cbz were protected from rot, reducing the sporulation index with respect to the control peaches (Figure 4a–c). Both treatments significantly reduced the brown rot sporulation index ($p < 0.05$) compared to the control peaches, with no significant differences between them (Figure 4d).

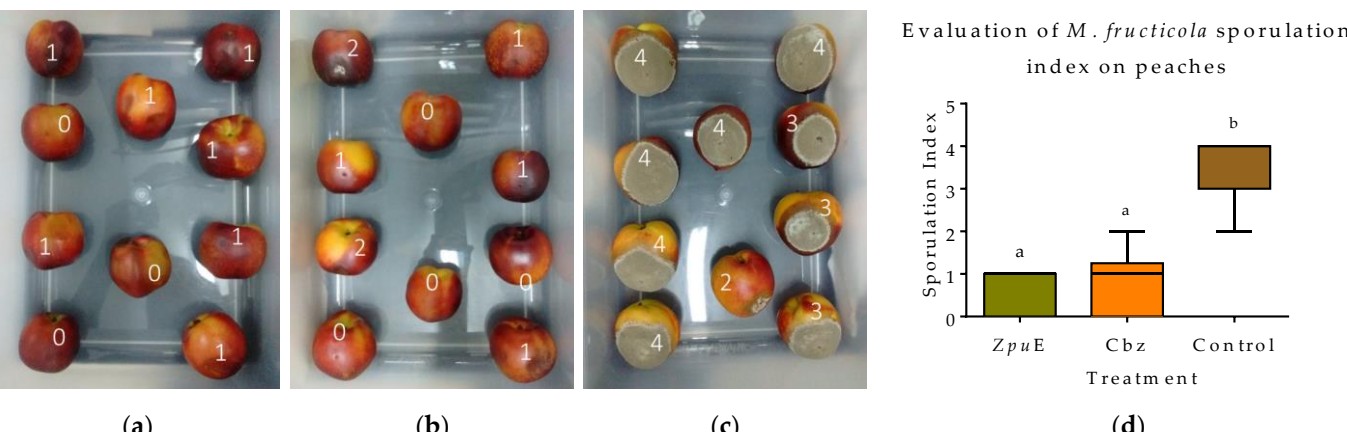

**Figure 4.** Pictures showing the *M. fructicola* sporulation index (see the number in each fruit) on wound-inoculated peaches treated with (**a**) *Z. punctata* ethanolic extract (*Zpu*E), (**b**) commercial Cbz, and (**c**) the control set without any treatment. (**d**) Data statistical analysis; different letters indicate statistically significant differences between treatments in accordance with Kruskal-Wallis multiple comparison tests ($p < 0.05$).

Furthermore, when examining the total set of treated peaches, it became evident that while some specimens showed clear signs of brown rot, fewer fruits exposed to the

extract appeared to be infected. In addition, although there were no significant differences between fruits treated with Cbz and those treated with the extract (Figure 4d), the results suggested that *Zpu*E would be even more potent than the commercial fungicide under the conditions tested.

### 3.5. Cytotoxicity Evaluation of Ethanolic Extract of Z. punctata Compared with Commercial Fungicides

The cell viability assay is one of the most important parameters for toxicological studies to evaluate how human cells react when exposed to agrochemicals and to identify possible toxic or harmful effects that may compromise human health, especially in the case of postharvest fungicides.

The viability of Huh7 cells was evaluated in vitro in the presence of *Zpu*E, Cbz, and Cap at different concentrations, in accordance with those used in the ex vivo assay (Figure 5).

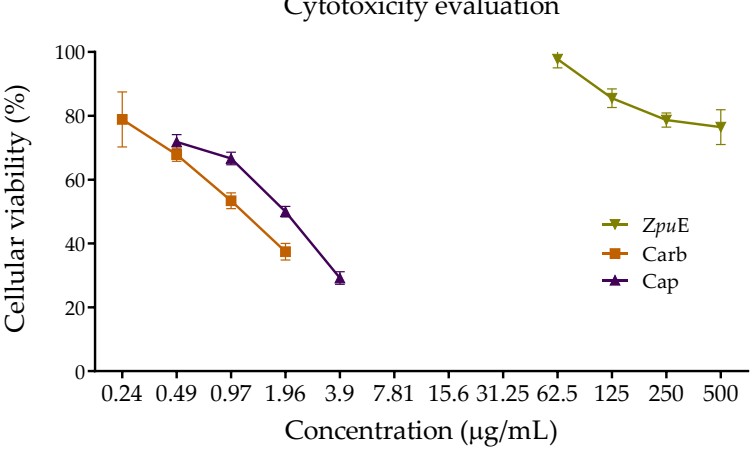

**Figure 5.** Viability of Huh7 cells in direct contact with different concentrations of the ethanolic extract of *Z. punctata* (*Zpu*E) and the commercial fungicides Carbendazim (Cbz) and Captan (Cap). Concentrations ranged from 2 MIC to 1/4 MIC of each sample, with MIC being the concentration used in the ex vivo assay (*Zpu*E = 250 μg/mL; Cbz = 0.97 μg/mL; Cap = 1.9 μg/mL). Values were calculated as means with standard deviation, determined by sextuplicate.

The viability of Huh7 cells in the presence of *Zpu*E, Cbz, and Cap at their MICs (*Zpu*E = 250 μg/mL; Cbz = 0.97 μg/mL; Cap = 1.9 μg/mL)., obtained in the broth microdilution assay and used in the ex vivo assay, was 78.69 ± 1.06%, 53.38 ± 1.92%, and 49.98 ± 1.20%, respectively. This means that at the effective concentration that inhibits fungal growth, *Zpu*E is considerably less cytotoxic to Huh7 cells than commercial fungicides, since higher cell viability values indicate lower cytotoxicity. As expected, the highest cell viability was observed in the most diluted solutions of each sample, corresponding to 1/4 MIC (*Zpu*E = 62.5 μg/mL; Cbz = 0.24 μg/mL, and Cap = 0.49 μg/mL).

## 4. Discussion

Food fungal infections cause significant global economic loss. Although synthetic fungicides have historically helped maintain production yields, their repeated application has become costly, and consumers are increasingly concerned about toxic residues in fruits and the environment. Moreover, the overuse of these chemicals has led to fungal resistance and risks to the health of those applying them, necessitating the search for alternative strategies.

Plant extracts with antifungal properties have gained increasing importance in this context. Numerous studies have explored the in vitro and in vivo antifungal activity of these extracts [53]. They are valuable because of the presence of multiple antifungal substances with diverse modes of action that can work in combination or synergistically to prevent or reduce microbial resistance [54]. Furthermore, if these plant metabolites

are not harmful to human and animal health and promote environmental sustainability, they represent excellent alternatives for controlling phytopathogenic fungi. In addition, the production process for these extracts is relatively simple and user-friendly, making them a practical solution in agricultural settings [55].

Some examples of the use of plant extracts to control postharvest infections in fruits are described below. Application of the aqueous extract of *Dianthus caryophyllus* to mango fruit showed a marked decrease in disease severity against *Alternaria alternata* [56]. The filtered fresh extract of *Aloe vera* effectively retarded the mycelial growth of four pathogenic fungi on papaya after 72 h of inoculation [57]. Pomegranate peel extract, despite having lower activity in in vitro assays, was effective in prolonging the shelf life of fruits against the pathogens *A. alternata* and *Penicillium expansum* on apples and pears [58].

The native herb *Z. punctata* has well-documented antifungal properties against dermatophytes and crop-pathogenic fungi [26]. Based on this knowledge, we focused our investigation on controlling fruit phytopathogenic fungi with the aim of applying active extracts to fresh fruit experiments. We demonstrated that *Zpu*E was active against *M. fructicola* (MIC/MFC = 250 µg/mL), which is consistent with previous findings on its efficacy against various plant pathogens [21,25–29]. Importantly, this result is significant, as many plant extracts showing high in vitro anti-fungal activity often fall short of in vivo assays [59]. Therefore, this plant is a promising source of broad-spectrum active metabolites in diverse fungal species.

Chalcones 1 and 2 were isolated from the extract, showing activity against *M. fructicola*, and through the bioautography assay it was demonstrated that these compounds were the main responsible for the ethanolic extract antifungal activity. Although the MICs and MFCs (between 62.5 and 125 µg/mL) were higher than those observed against previously evaluated crop pathogens (MICs and MFCs between 3.12 and 50 µg/mL) [26], this filamentous fungus is much more difficult to eradicate; therefore, these results are encouraging.

Motivated by our previous work on fresh fruits [60] and to promote full-spectrum extracts rather than pure compounds, the activity of *Zpu*E (containing 0.24 g of chalcone 1 and 0.12 g of chalcone 2/g of dry extract) was tested ex vivo on freshly collected peaches, demonstrating that, under the experimental conditions, the results of applying the plant extract were not statistically different from those obtained by exposing the fruits to Cbz.

Surprisingly, the primary cytotoxic experiments yielded results indicating significantly lower toxicity of *Zpu*E than Cbz and Cap at the concentrations tested in the ex vivo assay (MIC). Generally, the cell viability curve of *Zpu*E at the four tested concentrations was clearly more favorable than that of the two fungicides. At the highest concentration of *Zpu*E tested (2 MIC = 500 µg/mL), the cell viability value (76.44 $\pm$ 3.21%) was similar to that of the most diluted samples (1/4 MIC) of Cbz and Cap (0.24 µg/mL and 0.49 µg/mL, respectively). At these concentrations, cell viability was 78.86 $\pm$ 5.95% and 71.81 $\pm$ 1.1%, respectively. Although the effective concentration (MIC) of *Zpu*E used in the ex vivo experiments was much higher than that of commercial fungicides, it is worth noting the low toxicity of this natural complex mixture compared to pure synthetic fungicides, which are highly toxic even at very low concentrations.

These findings underscore the potential of plant extracts as a safer and environmentally friendly alternative for controlling phytopathogenic fungi. The reduced cytotoxicity of the extract not only suggests its suitability for agricultural applications but also raises the prospect of mitigating potential health and environmental risks associated with the use of conventional commercial fungicides. This observation further highlights the importance of pursuing natural and sustainable approaches to pest and disease management in agriculture with the potential to enhance both food safety and ecological integrity.

## 5. Conclusions

Finding potent and economic alternatives to current fungicides for controlling postharvest phytopathogenic fungi is indispensable. Following our hypothesis that the Argentinean medicinal plant *Zuccagnia punctata* (Fabaceae), which is extensively used for the

treatment of fungal and bacterial infections in human beings, may provide a solution to this case. We selected as a target a well-characterized pathogenic and virulent strain of *Monilinia fructicola*, which is the most hostile species of the genus that causes high economic losses in peach production worldwide. Consequently, we explored the in vitro antifungal activity of the ethanolic extract of *Z. punctata* against this phytopathogen. The chalcones 2′,4′-dihydroxy-3′-methoxychalcone and 2′,4′-dihydroxychalcone were isolated from the extract and evaluated against *M. fructicola* revealing that the observed biological activity was strongly related to the presence of these compounds in the extract. The chalcones were characterized by spectroscopic means that revealed their identity in comparison with previous literature.

In order to promote full-spectrum extract rather than pure compounds, the activity of the full extract at 250 µg/mL concentration was tested ex vivo on freshly collected peaches, demonstrating that under the experimental conditions, the results of applying the *Z. punctata* ethanolic extract were not statistically different than those obtained from exposing the fruits to the commercial product like Cbz and Cap. The overall results are promising and suggest that the two chalcones isolated from the plant as well as the full ethanolic extract may have great potential to be further developed as usable protective-contact fungicides.

In addition, an in vitro test to observe the viability of human hepatoma cells was performed in order to compare the cytotoxicity of our natural product and the fungicides currently and massively used for treating peach brown rot. Cytotoxicity evaluation showed that *Z. punctata* ethanolic extract was genuinely less cytotoxic than the commercial fungicides Cbz and Cap at the fungicidal concentrations (250, 0.97, and 1.9 µg/mL respectively), concluding that this plant extract could be potentially used to treat the peach disease as a more safety strategy than commercial fungicides. Finally, we wish to emphasize the importance of the research into natural fungicidal products that provide greater food and environmental security. Although the effective doses of natural products may be slightly higher than the doses of commercial fungicides, the quality of fruit that would be offered to the consumer will be highly appreciated.

**Supplementary Materials:** The following supporting information can be downloaded at: https://www.mdpi.com/article/10.3390/horticulturae9101141/s1, Figure S1: [1]H NMR spectrum of **1** in CDCl$_3$; Figure S2: [13]C NMR spectrum of **1**; Figure S3: EI mass spectrum of **1**; Figure S4: [1]H NMR spectrum of **2**; Figure S5: [13]C NMR spectrum of **2**; Figure S6: EI mass spectrum of **2**.

**Author Contributions:** Conceptualization, M.G.D. and L.A.S.; methodology, M.G.D.L., L.N.F. and M.I.S.; software, M.G.D.L., L.N.F. and M.I.S.; validation, M.G.D. and L.A.S.; formal analysis, M.G.D.L., L.N.F., L.A.S. and M.G.D.; investigation, M.G.D.L., M.I.S. and A.D.Q.; resources, L.A.S. and M.G.D.; data curation, L.N.F., L.A.S. and M.G.D.; writing—original draft preparation, M.G.D.L., L.N.F., A.D.Q. and M.I.S.; writing—review and editing, L.A.S. and M.G.D.; visualization, L.A.S. and M.G.D.; supervision, L.A.S. and M.G.D.; project administration, L.A.S. and M.G.D.; funding acquisition, L.A.S. and M.G.D. All authors have read and agreed to the published version of the manuscript.

**Funding:** This research was funded by Agencia Nacional de Promoción Científica Y Tecnológica (ANPCyT), grant numbers PICT-2020-SERIEA-02504, PICT-2021-CAT-II-00097, PICT 2021-267 and PICT-2020-SERIEA-02687; and by Consejo Nacional de Investigaciones Científicas y Técnicas (CONICET) under grant code PIP 11220210100388CO. The APC was funded by Consejo Nacional de Investigaciones Científicas y Técnicas (CONICET) under grant code PIP 11220210100388CO.

**Data Availability Statement:** Not applicable.

**Acknowledgments:** M.G.D.L., M.I.S., and L.N.F. thank CONICET for their scholarships. The authors wish to acknowledge the technical support received by María Alejandra Favaro and Mauro Sebastián Alisio; and complementary financial support from Agencia Santafesina de Ciencia, Tecnología e Innovación (ASaCTeI) and Universidad Nacional de Rosario (UNR) under project 80020190400002UR.

**Conflicts of Interest:** The authors declare no conflict of interest. The funders had no role in the design of the study; in the collection, analyses, or interpretation of data; in the writing of the manuscript; or in the decision to publish the results.

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
