# Peer review of "Control of Brown Rot Produced by Monilinia fructicola in Peaches Using a Full-Spectrum Extract of Zuccagnia punctata Cav."

_horticulturae, doi:10.3390/horticulturae9101141_

Round 1

Reviewer 1 Report

Comments and Suggestions for Authors

This research article investigated the antifungal activity of the ethanolic crude extract of Zuccagnia punctata against Monilinia fructicola in peaches. The results proved the potential of this extract and two isolated chalcones to inhibit the phytopathogen. In addition, the extract showed less toxicity on peaches than synthetic antifungals proving its potential as a natural alternative in agricultural practices.

Overall, the manuscript is well-written. 

Minor comments :

Line 90 - 92: this part should only state the aim of the work and not give the results. Please eliminate this idea.

Line 181 : The title "General" is vague and does not reflect the content of the section. Please replace it with a more appropriate one (Example: Peaches collection and preparation)

Line 265: bioatography > bioautography

Author Response

Responses to Reviewer 1:

Reviewer Comments: This research article investigated the antifungal activity of the ethanolic crude extract of Zuccagnia punctata against Monilinia fructicola in peaches. The results proved the potential of this extract and two isolated chalcones to inhibit the phytopathogen. In addition, the extract showed less toxicity on peaches than synthetic antifungals proving its potential as a natural alternative in agricultural practices. Overall, the manuscript is well-written. 

Response: Thank you very much for your kind words regarding our MS.

Reviewer minor comments:

Line 90 - 92: this part should only state the aim of the work and not give the results. Please eliminate this idea.

Response: this idea has been eliminated.

Line 181: The title "General" is vague and does not reflect the content of the section. Please replace it with a more appropriate one (Example: Peaches collection and preparation).

Response: The title has been replaced by the suggested one.

Line 265: bioatography > bioautography

Response: corrected.

Reviewer 2 Report

Comments and Suggestions for Authors

Paper is presenting very  interesting activity of extracts from Z. punctata against some phytopatogens. 

1. I advice to increase size of  chemical structures of 2’,4’-dihydroxy-3’-methoxychalcone and 2’,4’-dihydroxychalcone which were isolated from  aerial parts of Z. punctata; Also move  letter "alpha and beta" from double bond

2. Unify convention of units (I mean  mL or ml);

3. Paper " Svetaz, L., Tapia, A., López, S. N., Furlán, R. L., Petenatti, E., Pioli, R., ... & Zacchino, S. A. (2004). Antifungal chalcones and new caffeic acid esters from Zuccagnia punctata acting against soybean infecting fungi. Journal of Agricultural and Food chemistry52(11), 3297-3300." is not open access. I recommend to add to supplementary materials 1H and 13 NMR spectra. 

4. 1H multiplicuty (I mean J) are different due to frequency of NMR magnet. Did authors used 400 MHz as was in paper nr 26? Did authors used the same solvent also?

5. It's not clear to me, why more polar compound (I mean nr 1, presence of extra OMe group) is higher on TLC plate. Please re-check it. According to my experience, nr 1 should be more polar, so with lower Rf value;

Author Response

Responses to Reviewer 2:

Reviewer Comments: the paper presents a very interesting activity of extracts from Z. punctata against some phytopathogens. 

Reviewer: I advise increasing the size of chemical structures of 2’,4’-dihydroxy-3’-methoxychalcone and 2’,4’-dihydroxychalcone which were isolated from aerial parts of Z. punctata; Also move letters "alpha and beta" from the double bond.

Response: The size of chemical structures has been increased and alpha and beta letters were removed.

Reviewer: Unify convention of units (I mean mL or ml).

Response: The convention of units has been unified (mL).

Reviewer: Paper " Svetaz, L., Tapia, A., López, S. N., Furlán, R. L., Petenatti, E., Pioli, R., ... & Zacchino, S. A. (2004). Antifungal chalcones and new caffeic acid esters from Zuccagnia punctata acting against soybean infecting fungi. Journal of Agricultural and Food Chemistry, 52(11), 3297-3300." is not open access. I recommend adding to supplementary materials 1H and 13C NMR spectra. 

Response: 1H and 13C NMR spectra were added as supplementary materials. Moreover, we added another reference [41] that first described the chalcones' isolation and identification.

Reviewer: 1H multiplicity (I mean J) is different due to the frequency of the NMR magnet. Did the authors use 400 MHz as was in paper nr 26? Did the authors use the same solvent?

Response: We used the same conditions mentioned in reference [41] (300 MHz and the solvent CDCl3).

Reviewer: It's not clear to me, why a more polar compound (I mean nr 1, presence of extra OMe group) is higher on the TLC plate. Please re-check it. According to my experience, nr 1 should be more polar with a lower Rf value.

Response: compound 1 is higher on the TLC plate, despite the presence of an additional OMe group. The OMe group is adjacent to the OH groups, favoring the formation of H-bridges and detracting from the binding capacity of the molecule to the stationary phase, displacing more on the plate. Additionally, the elution solvent (Chloroform/ Ethyl acetate) favored the formation of this type of bond.

Reviewer 3 Report

Comments and Suggestions for Authors

Comments on the Quality of English Language

No

Author Response

Responses to Reviewer 3:

Reviewer Comments: The in vitro antifungal activity of the ethanolic extract of Zuccagnia punctata against the phytopathogen Monilinia fructicola was studied. Because the study is based on some results of predecessors, some of the content seems a little rough or redundant.

Reviewer: Table 1 can be deleted, and it can be described directly in a short text.

Response: Table 1 has been deleted and described in the text.

Reviewer: In Table 2, M. fructicola can be removed.

Response: M. fructicola has been removed.  

Reviewer: Figure 1c is not pretty, it is recommended to replace something more convincing.

Response: Figure 1c has been replaced.

Reviewer: Figure 2 can be deleted because there have been some previous reports (following two references).

Response: We consider it interesting to keep Figure 2 to facilitate the comprehension of future readers of this paper. But we put into consideration this suggestion to the editor.

Reviewer: Figure 3. There are many black shadows around the perimeter of the figures, it is recommended to revise and delete the surrounding black shadows, so the figure can be more compact.

Response: The black shadows around the perimeter of the figure were deleted.

Reviewer: Please add the two references in the text: 1. Chahar, F. C., Alvarez, P. E., Zampini, C., María I. Isla, & Silvia Antonia Brandán. (2020). Experimental and DFT studies on 2′,4′-dihydroxychalcone, a product isolated from Zuccagnia punctata cav. (Fabaceae) medicinal plant. Journal of Molecular Structure, 1201, 127221. 2. Estefanía Butassi, Laura A. Svetaz, Juan J. Ivancovich, Gabriela E. Feresin, Alejandro Tapia, Susana A. Zacchino, Synergistic mutual potentiation of antifungal activity of Zuccagnia punctata Cav. and Larrea nitida Cav. extracts in clinical isolates of Candida albicans and Candida glabrata, Phytomedicine (2015), doi: 10.1016/j.phymed.2015.04.004.

Response: These two references were added [40, 31].

Reviewer 4 Report

Comments and Suggestions for Authors

The Manuscript ID horticulturae-2625449 describes the antifungal activity of the ethanolic extract of Zuccagnia punctata and two derived chalcones against the phytopathogen Monilinia fructicola. The manuscript is interesting and has relevant results for readers. However, some points should be addressed prior to further consideration.

1.       Abstract: A conclusive sentence is missing in the abstract ending.

2.       Lines 36-74: These paragraphs mention only an extensive topic, so they can be organized better and summarized since some passages could be merged to deliver a more straightforward message for readers.

3.       Lines 76-92: This paragraph can be subdivided into two paragraphs to expand the justification and background of the test botanical in one paragraph, and the aim & scope and the introductive experimental outcome in the second paragraph.

4.       Line 111: 1H and 13C NMR data and spectra must be provided as supplementary material.

5.       Line 145: The deposit or accession number should be provided here instead of in the Results section.

6.       Line 185: The agronomic regime for the plants to obtain the test fruits must be mentioned.

7.       Section 3.1 and Table 1: This section is partially mentioned in the M&M section. It can be accordingly moved.

8.       Figure 1 is not adequately explained and described in the text and should be expanded.

9.       Line 273: Compound with Rf = 0.28 should be elucidated, and its results should be added to this manuscript.

10.   Figure 5: The X axis of this figure must include concentrations instead of dilutions since it is very confusing for readers, and the relationship with MIC will be more precise.

11.   The results section must be improved and even rewritten since several explanations and descriptions are missing or insufficiently mentioned.

12.   The discussion should be improved since it is highly descriptive and provides marginal comparative ideas with other studies.

13.   Although it is not mandatory, a conclusion section is missing. It should be included with conceptual findings from a mechanistic point of view and even the scope of these results.

Comments on the Quality of English Language

Detailed scrutiny should be performed throughout the manuscript to revise/correct several grammar and stylistic issues since some passages are challenging to be followed.

Author Response

Responses to Reviewer 4:

Reviewer Comments: The Manuscript ID horticulturae-2625449 describes the antifungal activity of the ethanolic extract of Zuccagnia punctata and two derived chalcones against the phytopathogen Monilinia fructicola. The manuscript is interesting and has relevant results for readers. However, some points should be addressed prior to further consideration.

Reviewer: Abstract: A conclusive sentence is missing in the abstract ending.

Response: A conclusive sentence has been added.

Reviewer: Lines 36-74: These paragraphs mention only an extensive topic, so they can be organized better and summarized since some passages could be merged to deliver a more straightforward message for readers.

Response: These paragraphs were also reorganized based on another reviewer's feedback.

Reviewer: Lines 76-92: This paragraph can be subdivided into two paragraphs to expand the justification and background of the test botanical in one paragraph, and the aim & scope and the introductive experimental outcome in the second paragraph.

Response: This paragraph has been subdivided into two paragraphs according to this suggestion.

Reviewer: Line 111: 1H and 13C NMR data and spectra must be provided as supplementary material.

Response: 1H and 13C NMR data and spectra are now provided as supplementary material.

Reviewer: Line 145: The deposit or accession number should be provided here instead of in the Results section.

Response: the accession number has been provided here.

Reviewer: Line 185: The agronomic regime for the plants to obtain the test fruits must be mentioned.

Response: The main topic of this query has been mentioned in section 2.3.4.1.

Reviewer: Section 3.1 and Table 1: This section is partially mentioned in the M&M section. It can be accordingly moved.

Response: Section 3.1 and Table 1 were removed as suggested by another reviewer too.

Reviewer: Figure 1 is not adequately explained and described in the text and should be expanded.

Response: Figure 1 has been explained and described in more detail in the text.

Reviewer: Line 273: Compound with Rf = 0.28 should be elucidated, and its results should be added to this manuscript.

Response:

The aim of this study was to use a full-spectrum plant extract characterized by the two most active major compounds. It can be inferred from a previous work [29] that the inhibition halo at Rf = 0.28 could correspond to fungitoxic flavonoids present in the plant, such as pinocembrin or galangin, but their correct characterization and inclusion in the manuscript is beyond the scope of this work.

Reviewer: Figure 5: The X axis of this figure must include concentrations instead of dilutions since it is very confusing for readers, and the relationship with MIC will be more precise.

Response: Because the MIC of each component is different, we believe that the use of numerical concentration values would generate more confusion for readers. This is explained in the caption of the figure. In addition, the dilutions were replaced with the corresponding MIC dilutions.

Reviewer: The results section must be improved and even rewritten since several explanations and descriptions are missing or insufficiently mentioned.

Response: The results section has been improved. The explanations of some results in the discussion section have been expanded.

Reviewer: The discussion should be improved since it is highly descriptive and provides marginal comparative ideas with other studies.

Response: The discussion has been improved.

Reviewer: Although it is not mandatory, a conclusion section is missing. It should be included with conceptual findings from a mechanistic point of view and even the scope of these results.

Response: A conclusion section has been added.

Reviewer 5 Report

Comments and Suggestions for Authors

The current manuscript entitled "Control of brown rot produced by Monilinia fructicola in peaches using a full-spectrum extract of Zuccagnia punctata Cav.” investigated the isolation a pathogenic strain of Monilinia fructicola, which is the most aggressive species within its genus. The in vitro antifungal properties of an ethanolic extract of Z. punctata against this destructive phytopathogen was investigated. The identification of two key chalcones, 2’,4’-dihydroxy-3’-methoxychalcone and 2’,4’-dihydroxychalcone was found within the extract. Authors suggested that these compounds were shown to be responsible for the observed antifungal activity against M. fructicola. Then, ex-vivo experiment involving fresh peaches that had been inoculated with the pathogen was conducted to test the activity of Z. punctata extract as antifungal compared to 2 different  commercial fungicides. The cytotoxicity test on Huh7 cells was also conducted.

Comments:

Introduction section, line 89-92 included comments on the obtained results. These sentences should be removed from the introduction and moved to the result section.

Results section, 3.1. Plant Collection and Extract Preparation. All information on that part should be presented within the materials and methods section. Table 1 is not significant; authors can mention within the text that ethanol extract production yield was 27.7 g.

Line 258-262, these sentences should be moved to the discussion section.

How the M. fructicola sporulation index was calculated?

The Discussion section should be expanded to provide more specific explanations of the obtained results, incorporating relevant comparisons to more recent references.

The Conclusion section should be added to emphasize the key findings of the current study.

Comments on the Quality of English Language

Moderate revisions are needed to improve the English language.

Author Response

Responses to Reviewer 5:

Reviewer Comments: The current manuscript entitled "Control of brown rot produced by Monilinia fructicola in peaches using a full-spectrum extract of Zuccagnia punctata Cav.” investigated the isolation of a pathogenic strain of Monilinia fructicola, which is the most aggressive species within its genus. The in vitro antifungal properties of an ethanolic extract of Z. punctata against this destructive phytopathogen were investigated. The identification of two key chalcones, 2’,4’-dihydroxy-3’-methoxychalcone and 2’,4’-dihydroxychalcone was found within the extract. Authors suggested that these compounds were shown to be responsible for the observed antifungal activity against M. fructicola. Then, an ex-vivo experiment involving fresh peaches that had been inoculated with the pathogen was conducted to test the activity of Z. punctata extract as an antifungal compared to 2 different commercial fungicides. The cytotoxicity test on Huh7 cells was also conducted.

Reviewer: introduction section, lines 89-92 included comments on the obtained results. These sentences should be removed from the introduction and moved to the result section.

Response: These sentences were moved to the results section.

Reviewer: results section, 3.1. Plant Collection and Extract Preparation. All information on that part should be presented within the materials and methods section. Table 1 is not significant; the authors can mention within the text that the ethanol extract production yield was 27.7 g.

Response: This section has been deleted because it was presented in the material and methods section.

Reviewer: Lines 258-262, these sentences should be moved to the discussion section.

Response: This sentence has been moved to the discussion section.

Reviewer: how the M. fructicola sporulation index was calculated?

Response: This is explained in section 2.3.4.3.

Reviewer: The discussion section should be expanded to provide more specific explanations of the obtained results, incorporating relevant comparisons to more recent references.

Response: The discussion section has been expanded.

Reviewer: The Conclusion section should be added to emphasize the key findings of the current study.

Response: A conclusion section has been added.

Reviewer 6 Report

Comments and Suggestions for Authors

"Control of brown rot produced by Monilinia fructicola in peaches using a full-spectrum extract of Zuccagnia punctata Cav." is the title of the proposed article. Authors explored the in vitro antifungal activity of the ethanolic extract of Z. punctata and 2’,4’-dihydroxy-3’-methoxychalcone and 2’,4’-dihydroxychalcone against M. fructicola. Also, ex vivo assays on fresh peaches were carried out. Final step was a cytotoxicity assays on Human Hepatoma cells, authors showed that ethanol extract was less cytotoxic than commercial fungicides at the assayed concentrations.

The manuscript is clearly written, the results and discussion are written in a concise manner that can only apparently be considered negatively. In this reviewer's opinion this is a positive aspect. The bibliography is sufficient and is consistent with the concise nature of the work. There are only a few flaws that need to be improved before acceptance.

Points that need to be addressed:

Line 71: Please, probably it is better to put the citation number just after the author's name.

Figure 1: Please, authors must check bold letters on panels and legend.

Figure 3: Please, see figure 1.

Figure 4: Please, authors must improve the quality of the figure 4d. It is something like a b/w slide. Check the bold letters on figure.

Figure 5: Please the author to improve the quality of the figure, the size of the points and the standard deviation symbols can sometimes cause confusion for the reader.

Discussion: It is true that the synthetic nature has been praised previously... I believe that greater depth in the discussion should be done on the lower toxicity of the natural product compared to pesticides. Two lines are not enough compared to the importance of the data. I recommend seeing some other examples, possible connections and future applications.

Author Response

Responses to Reviewer 6:

Reviewer Comments: Control of brown rot produced by Monilinia fructicola in peaches using a full-spectrum extract of Zuccagnia punctata Cav." is the title of the proposed article. Authors explored the in vitro antifungal activity of the ethanolic extract of Z. punctata and 2’,4’-dihydroxy-3’-methoxychalcone and 2’,4’-dihydroxychalcone against M. fructicola. Also, ex vivo assays on fresh peaches were carried out. The final step was a cytotoxicity assays on Human Hepatoma cells, authors showed that ethanol extract was less cytotoxic than commercial fungicides at the assayed concentrations.

The manuscript is clearly written, and the results and discussion are written in a concise manner that can only apparently be considered negatively. In this reviewer's opinion, this is a positive aspect. The bibliography is sufficient and is consistent with the concise nature of the work. There are only a few flaws that need to be improved before acceptance.

Response: Thank you very much for your kind words about our work.

Points that need to be addressed:

Reviewer: Line 71: Please, probably it is better to put the citation number just after the author's name.

Response: This part of the introduction has been changed but the reference was kept only with the citation number.

Reviewer: Figure 1: Please, authors must check bold letters on panels and legends.

Response: Bold letters have been normalized in Figure 1.

Reviewer: Figure 3: Please, see Figure 1.

Response: Bold letters were normalized.

Reviewer: Figure 4: Please, authors must improve the quality of the figure 4d. It is something like a b/w slide. Check the bold letters on the figure.

Response: The quality of Figure 4d has been improved and bold letters normalized.

Reviewer: Figure 5: Please the author to improve the quality of the figure, the size of the points and the standard deviation symbols can sometimes cause confusion for the reader.

Response: Figure 5 has been improved.

Reviewer: Discussion: It is true that the synthetic nature has been praised previously... I believe that greater depth in the discussion should be done on the lower toxicity of natural products compared to pesticides. Two lines are not enough compared to the importance of the data. I recommend seeing some other examples, possible connections, and future applications.

Response: This has been emphasized in the discussion and conclusion sections.

Round 2

Reviewer 3 Report

Comments and Suggestions for Authors

No comment.

Author Response

Thank you veru much for your revision

Reviewer 4 Report

Comments and Suggestions for Authors

The authors addressed the main body of my comments, so the manuscript improved substantially. However, I'm afraid I have to disagree with some responses regarding my comments, as following described:

1. Authors' Response: The aim of this study was to use a full-spectrum plant extract characterized by the two most active major compounds. It can be inferred from a previous work [29] that the inhibition halo at Rf = 0.28 could correspond to fungitoxic flavonoids present in the plant, such as pinocembrin or galangin, but their correct characterization and inclusion in the manuscript is beyond the scope of this work. 

Derived Comment: Why beyond this study's scope? If I understood well, according to subheading 3.3., a study's focus was oriented to detect the bioactive compounds in the extract by bioautography, and the compound with Rf 0.28 is bioactive! (Figure 3e!). So, the authors' reason mentioned in this response does not sound entirely logical. Therefore, I would insist on including this compound in this study to complete such a study's objective. However, considering that some logistical/technical issues could limit such an inclusion, I suggest expanding the real reason (in lines 314-315) and explaining correctly and clearly why this active compound was not completely characterized and included in the subsequent bioassays. This extended explanation is crucial for the manuscript and to deliver a piece of transparent information to the readership. 

2. Authors' Response: Because the MIC of each component is different, we believe that the use of numerical concentration values would generate more confusion for readers. This is explained in the caption of the figure. In addition, the dilutions were replaced with the corresponding MIC dilutions.

Derived Comment: Sure. The MIC of each test agent is different, and this is the main concern related to my comment!. Such a presentation will be very confusing for readers and may even contribute to erroneous interpretations in the future since the MICs of test substances are not the same, but the authors directly compared the dose-dependent response (i.e., cell viability) using a different parameter (i.e., MIC) that implies different concentrations but they are depicted as the same within a line data plot. In other words, a comparable response behavior with different concentrations has no sense in a dose-response curve. Contrarily, if concentrations are provided, a correct comparison can adequately be made, and sure, the curve trend will be shifted to distinct ranges within the X-axis (dose), but such a presentation will be more beneficial to see the scope of the extract effect in comparison with the commercial fungicides. Perhaps a decadic logarithmic scale will be beneficial to such a purpose. However, if the authors prefer to retain their idea by relating the response to the MIC proportions (be aware in this case that they are not entirely related to dilutions since 2-fold MIC (2MIC) is not a dilution of the reference value MIC), which could be valid, the main problem is on using a line data plot since it could be misinterpreted as a dose-response curve, and it is thus not correct. So, a bar graph or, even better, a box plot (to see the data distribution and dispersion) is recommended instead of a line plot. Therefore, I recommend changing the presentation of this figure 5 since it constitutes a scientific flaw.

In addition, but not less importantly, the authors must modify the text regarding the Figure 5-derived explanations. For instance, in line 354, the authors mentioned "different concentrations," but this term needs to be clarified if it is related to the different concentrations regarding MICs (implying different concentrations) or related to the tested four different MIC proportions. Additionally, revise the passage in line 336: if MIC involves different concentrations, it is incorrect to indicate that a treatment is less toxic than others, considering that the cell viability depends on the dose, and the authors assessed different MIC-related concentrations. I insist, therefore, that the comparison using MIC proportions is highly confusing. I suggest revising, improving, and being careful with the explanations derived from Figure 5.

3. Authors' response: A conclusion section has been added.

Derived comment: Although the authors added a conclusions section, this section summarizes results and contains a marginal outlook, but conceptual findings from a mechanistic point of view are not provided.

Comments on the Quality of English Language

There are still several style and grammar issues throughout the manuscript and even typos. For instance, spectra instead of spectrums (line 141). So, a detailed and careful revision, possibly through an editing service, is then recommended. 

Author Response

Responses to Reviewer 4 (second round):

Thank you very much for your revision and suggestions.

Reviewer Comments and Suggestions for Authors: The authors addressed the main body of my comments, so the manuscript improved substantially. However, I'm afraid I have to disagree with some responses regarding my comments, as following described:

  1. Authors' Response: The aim of this study was to use a full-spectrum plant extract characterized by the two most active major compounds. It can be inferred from a previous work [29] that the inhibition halo at Rf = 0.28 could correspond to fungitoxic flavonoids present in the plant, such as pinocembrin or galangin, but their correct characterization and inclusion in the manuscript is beyond the scope of this work. 

Derived Comment: Why beyond this study's scope? If I understood well, according to subheading 3.3., a study's focus was oriented to detect the bioactive compounds in the extract by bioautography, and the compound with Rf 0.28 is bioactive! (Figure 3e!). So, the authors' reason mentioned in this response does not sound entirely logical. Therefore, I would insist on including this compound in this study to complete such a study's objective. However, considering that some logistical/technical issues could limit such an inclusion, I suggest expanding the real reason (in lines 314-315) and explaining correctly and clearly why this active compound was not completely characterized and included in the subsequent bioassays. This extended explanation is crucial for the manuscript and to deliver a piece of transparent information to the readership. 

Response: The reason why the compound(s) were not characterized is now included in section 3.3. The subheading was also changed.

  1. Authors' Response: Because the MIC of each component is different, we believe that the use of numerical concentration values would generate more confusion for readers. This is explained in the caption of the figure. In addition, the dilutions were replaced with the corresponding MIC dilutions.

Derived Comment: Sure. The MIC of each test agent is different, and this is the main concern related to my comment!. Such a presentation will be very confusing for readers and may even contribute to erroneous interpretations in the future since the MICs of test substances are not the same, but the authors directly compared the dose-dependent response (i.e., cell viability) using a different parameter (i.e., MIC) that implies different concentrations but they are depicted as the same within a line data plot. In other words, a comparable response behavior with different concentrations has no sense in a dose-response curve. Contrarily, if concentrations are provided, a correct comparison can adequately be made, and sure, the curve trend will be shifted to distinct ranges within the X-axis (dose), but such a presentation will be more beneficial to see the scope of the extract effect in comparison with the commercial fungicides. Perhaps a decadic logarithmic scale will be beneficial to such a purpose. However, if the authors prefer to retain their idea by relating the response to the MIC proportions (be aware in this case that they are not entirely related to dilutions since 2-fold MIC (2MIC) is not a dilution of the reference value MIC), which could be valid, the main problem is on using a line data plot since it could be misinterpreted as a dose-response curve, and it is thus not correct. So, a bar graph or, even better, a box plot (to see the data distribution and dispersion) is recommended instead of a line plot. Therefore, I recommend changing the presentation of this figure 5 since it constitutes a scientific flaw.

In addition, but not less importantly, the authors must modify the text regarding the Figure 5-derived explanations. For instance, in line 354, the authors mentioned "different concentrations," but this term needs to be clarified if it is related to the different concentrations regarding MICs (implying different concentrations) or related to the tested four different MIC proportions. Additionally, revise the passage in line 336: if MIC involves different concentrations, it is incorrect to indicate that a treatment is less toxic than others, considering that the cell viability depends on the dose, and the authors assessed different MIC-related concentrations. I insist, therefore, that the comparison using MIC proportions is highly confusing. I suggest revising, improving, and being careful with the explanations derived from Figure 5.

Response: The figure was changed to include the concentrations used in the assay. The explanations in the results and discussion sections were also expanded. We wanted to demonstrate in this work that at the dose at which the inhibition of fungal growth occurs, i.e. at the MIC, the extract is not toxic compared to commercial fungicides. That is, at the same effect, the less toxicity.

  1. Authors' response: A conclusion section has been added.

Derived commentAlthough the authors added a conclusions section, this section summarizes results and contains a marginal outlook, but conceptual findings from a mechanistic point of view are not provided.

Response: we did our best to improve this section.

Reviewer 5 Report

Comments and Suggestions for Authors

The manuscript was improved following the authors' thorough revisions and corrections. They addressed the majority of my comments, resulting in a current version that is well-prepared for publication.

Author Response

Thank you very much for your revision

Reviewer 6 Report

Comments and Suggestions for Authors

The authors fully answered to the previous remarks, the manuscript can be published

Author Response

Thank you very much for your revision